# New estimation of critical insolation – $CO_2$ relationship for triggering glacial inception

Stefanie Talento[1], Matteo Willeit[1], and Andrey Ganopolski[1]

[1]Potsdam Institute for Climate Impact Research (PIK), Member of the Leibniz Association, P.O. Box 601203, D-14412 Potsdam Germany

**Correspondence:** Matteo Willeit (willeit@pik-potsdam.de)

**Abstract.** It has been previously proposed that glacial inception represents a bifurcation transition between interglacial and glacial states, and is governed by the non-linear dynamics of the climate-cryosphere system. To trigger glacial inception, the orbital forcing (defined as the maximum of summer insolation at 65oN and determined by Earth's orbital parameters) must be lower than a critical level, which depends on the atmospheric $CO_2$ concentration. While paleoclimatic data do not provide a
strong constraint on the dependence between $CO_2$ and critical insolation, its accurate estimation is of fundamental importance for predicting future glaciations and the effect that anthropogenic $CO_2$ emissions might have on them. In this study, we use the novel Earth system model of intermediate complexity CLIMBER-X with interactive ice sheets to produce a new estimation of the critical insolation - $CO_2$ relationship for triggering glacial inception. We perform a series of experiments in which different combinations of orbital forcing and atmospheric $CO_2$ concentration are maintained constant in time. We analyse for which
combinations of orbital forcing and $CO_2$ glacial inception occurs and trace the critical relationship between them, separating conditions under which glacial inception is possible from those where glacial inception is not materialised. We also provide a theoretical foundation for the proposed critical insolation – $CO_2$ relation. We find that the use of the maximum summer insolation at 65°N as a single metric for orbital forcing is adequate for tracing the glacial inception bifurcation. Moreover, we find that the temporal and spatial patterns of ice sheet growth during glacial inception are not always the same but depend on
the critical insolation and $CO_2$ level. The experiments evidence that during glacial inception, ice sheets grow mostly in North America, and only under low $CO_2$ conditions ice sheets are also formed over Scandinavia. The latter is associated with a weak Atlantic Meridional Overturning Circulation (AMOC) for low $CO_2$. We find that the strength of AMOC also affects the rate of ice sheet growth during glacial inception.

## 1 Introduction

Glacial cycles are the dominant mode of climatic variability over the last 2.7 million years. The timing of glaciations and deglaciations is primarily controlled by changes in Earth's orbital parameters through the modulation of the amount of solar radiation received in high latitudes of the Northern Hemisphere (NH) during boreal summer (Milankovitch, 1941; Ganopolski, 2024).

Using an Earth system model of intermediate complexity (EMIC), Calov et al. (2005) proposed that glacial inception represents a bifurcation transition from interglacial to glacial states of the Earth system. When summer insolation at high latitudes of the NH falls below a certain threshold, the interglacial state becomes unstable and ice sheet growth begins. The process is amplified through non-linear feedbacks, of which the snow and ice-albedo and the elevation feedbacks are the dominant ones. Eventually, large ice sheets develop over the North American and the Eurasian continents.

The threshold value for boreal summer insolation to trigger a glacial inception depends on the atmospheric concentration of greenhouse gases, among which $CO_2$ is, by far, the most important (Köhler et al., 2010). While paleoclimatic data do not constrain this dependence, its accurate estimation is of fundamental importance for predicting future glaciations and the effect that anthropogenic $CO_2$ emissions might have on them (Archer and Ganopolski, 2005; Talento and Ganopolski, 2021).

The first attempt at tracing the relationship between insolation and $CO_2$ for triggering glacial inception was performed in Archer and Ganopolski (2005) using the EMIC CLIMBER-2 (Petoukhov et al., 2000; Ganopolski et al., 2001). This first estimation was, however, rather crude as it was based only on a small set of relatively short experiments that did not allow for the detection of the bifurcation with sufficient accuracy. Ganopolski et al. (2016) updated the results also using CLIMBER-2 with a new methodology that allowed for increased accuracy. The authors proposed a functional relationship between summer maximum insolation at 65°N ($smx65$) and $CO_2$ for triggering glacial inception. The relationship describes the critical $smx65$ for the onset of glaciation as linearly dependent on the logarithm of $CO_2$: $smx65_{cr} = \alpha \ln(CO_2/CO_{2_0}) + \beta$, where $CO_2$ is the atmospheric $CO_2$ concentration, $CO_{2_0}$ is a reference atmospheric $CO_2$ value (the preindustrial value) and $\alpha$ and $\beta$ are empirical constants.

The shape of the dependency is in agreement with the facts that (i) the radiative forcing of $CO_2$ follows a logarithmic structure and (ii) that in CLIMBER-2 the temperature response to radiative forcing of $CO_2$ and orbital forcing is linear within the considered range.

The purpose of this work is to build upon the findings of Ganopolski et al. (2016) and generate a new estimation of the relationship between insolation and atmospheric $CO_2$ critical for triggering glacial inception, using the recently developed and more advanced CLIMBER-X model (Willeit et al., 2022, 2023, 2024). While CLIMBER-X is also an EMIC that shares with CLIMBER-2 the principles of computational efficiency and usage of a statistical-dynamical atmospheric model, it represents a significant development compared to the CLIMBER-2 model used in the previous studies (Ganopolski and Brovkin, 2017; Willeit et al., 2019). In particular CLIMBER-X has (i) a much higher horizontal resolution in the atmosphere and land models (5°x5° versus ∼50°x 10° in a longitude-latitude grid), (ii) a 3D ocean model, (iii) more internally consistent components and (iv) better treatment of individual processes (e.g.: surface energy balance, precipitation, radiative transfer, sea ice dynamics, photosynthesis, vegetation dynamics). Furthermore, we investigate the temporal and spatial patterns of ice sheet growth during glacial inception and the effects of changes in the Atlantic Meridional Overturning Circulation (AMOC) strength on the rate of ice sheets growth. We also analyse the suitability of using the $smx65$ as a single metric for orbital forcing. A theoretical derivation of the critical insolation – $CO_2$ relationship for triggering glacial inception is provided in Appendix A.

## 2 Model and experimental setup

### 2.1 Model description

We employ the new CLIMBER-X EMIC (Willeit et al., 2022, 2023, 2024). The model was specifically designed to simulate the Earth system evolution on timescales from decades to hundreds of thousands of years. The climate component of CLIMBER-X consists of: the 2.5-dimensional Semi-Empirical dynamical-Statistical Atmosphere Model (SESAM), the 3-dimensional frictional-geostrophic ocean model GOLDSTEIN, the dynamic-thermodynamic SImple Sea Ice Model (SISIM) and the land surface–vegetation model PALADYN. All the models from the climate core are discretized on a longitude-latitude grid with $5°x5°$ horizontal resolution and use a daily time-step. The ocean model has 23 unequally spaced vertical layers, with a $10\,\mathrm{m}$ top layer and layer thickness increasing with depth and reaching $500\,\mathrm{m}$ at the ocean bottom. CLIMBER-X incorporates the 3-dimensional thermomechanical ice sheet model SICOPOLIS (Greve et al., 2017) via the high-resolution physically-based surface energy and mass balance interface SEMIX (Willeit et al., 2024) and a basal ice shelf melt module. In this study, SICOPOLIS is applied only to the NH with a $32\,\mathrm{km}$ horizontal resolution. The viscoelastic solid Earth model VILMA (Klemann et al., 2008; Martinec et al., 2018) is used to simulate the bedrock response to changes in loading by solving the sea-level equation. A detailed description of the coupling between climate and ice sheet components is provided in Willeit et al. (2024). CLIMBER-X does not simulate internal decadal-scale variability by its design, but it can simulate the forced response at this time scale, such as response to volcanic eruptions and $CO_2$ changes (Willeit et al., 2022). The equilibrium climate sensitivity of CLIMBER-X is $\sim3\,°\mathrm{C}$, close to the best-guess estimate from the IPCC (IPCC 2021, 2021).

Even though CLIMBER-X has a relatively coarse spatial resolution and a simplified treatment of some processes, the number of physical processes included in the model, as well as its performance for the present-day climate, are comparable with state-of-the-art (CMIP6-type) climate models (Willeit et al., 2022, 2023). CLIMBER-X is reasonably successful in reproducing the present-day climatology of variables relevant for the formation of ice sheets. However, near-surface summer temperatures (a key variable for determining ablation of ice sheets) are up to $5\,°\mathrm{C}$ too warm over the eastern portion of North America when compared to observations (Willeit et al., 2024). Preliminary experiments show that such biases preclude the realistic development of North American ice-sheets. Therefore, similarly to Ganopolski et al. (2010), we implemented a 2m-temperature bias correction in the surface mass balance calculation (see Willeit et al. (2024). In addition, a uniform offset of -0.5 $°\mathrm{C}$ is applied globally in the surface mass balance scheme. As discussed in Willeit et al. (2024), the precipitation biases in CLIMBER-X are less problematic for the simulation of the surface mass balance of NH ice sheets.

### 2.1.1 Experimental setup

We perform a series of equilibrium experiments in which different orbital parameters (obliquity, eccentricity and precession), atmospheric $CO_2$ concentration, are maintained constant for the whole duration of the simulation. All NH ice sheets are fully interactive and influence climate through albedo, elevation, sea level (which affects land-sea mask) and freshwater flux into the ocean (Willeit et al., 2024). Since the ocean model in CLIMBER-X is based on the rigid-lid approximation we always enforce that the net annual global surface freshwater flux is zero. In the case of interactive ice sheets, we still enforce the global annual

surface freshwater flux to be zero, but then adjust the ocean volume each year to match the change in global ice volume (Willeit et al., 2022). The Antarctic Ice sheet is not interactive, but fixed at its present-day state. The experiments are performed using an acceleration technique with asynchronous coupling between the climate and ice sheet model components. An acceleration factor of 10 is used, i.e. the climate model component is updated every 10 years of the ice sheet model. If we were to calculate global ocean volume change by integrating surface freshwater fluxes, then a 10-fold acceleration would require us to aggregate these fluxes over 10 years, but since in our model the ocean volume change is not driven by surface freshwater fluxes this is not necessary and the freshwater fluxes are computed as in the non-accelerated case. Willeit et al. (2024) demonstrated that acceleration factors up to $\sim$10 are acceptable and provide realistic results of the last glacial inception. Each model simulation is run for 100.000 years of the ice sheet model (corresponding to 10.000 years of the climate model) starting from pre-industrial equilibrium conditions with observed present-day Greenland ice sheet with a uniform ice temperature of -10 °C.

We select 19 combinations of orbital parameters corresponding to the 19 local minima of the $smx65$ time-series of the last 400 kyr, corresponding to values in the range between 430 and 496 $Wm^{-2}$ (Fig. 1). Therefore, selected combinations of orbital parameters have similar precession but very different eccentricity and obliquity values. For each of these 19 sets of orbital parameters, we produce a series of equilibrium experiments designed to detect the critical insolation – $CO_2$ relationship for triggering inception. The procedure is, for each set of orbital parameters, as follows:

1. We start from a simulation using a (fairly high) atmospheric $CO_2$ concentration of 500 ppm, for which glacial inception is not expected to occur under any orbital configuration.

2. If the NH ice volume ($v$) is < $X$ msle (meters in sea level equivalent) during the whole simulation, we reduce $CO_2$ by 5 ppm and repeat the simulation.

If, on the contrary, $v$ is $\geq X$ msle at any moment in the simulation, we stop the iteration and define the insolation – $CO_2$ combination used in this case as critical for glacial inception.

In this study, we select a value of $X = 15$ msle. That value corresponds to an ice volume roughly twice the value of the present-day Greenland ice sheet volume. As in CLIMBER-X ice volume over Greenland does not increase substantially under glacial inception conditions, 15 msle is thus associated with a considerable ice sheet growth outside Greenland.

To validate the model suitability for tracing glacial inception bifurcation transitions, similarly to Ganopolski et al. (2016), we test the model performance against two paleoclimatic constraints. The first constraint is that, for present-day orbital configuration and atmospheric $CO_2$ concentration of 280 ppm (named Control simulation, Table 1), the model should not produce any substantial ice growth outside of Greenland. The second constraint is that, on the contrary, for the same 280 ppm level of atmospheric $CO_2$ concentration but an orbital configuration corresponding to the end of MIS11 (398 kyrBP) a significant ice growth, indicative of glacial inception, should occur. Under the current model version these two constraints are met: after 100 kyr of simulation the NH ice volume totals, including the Greenland ice sheet, 8.3 msle for the present-day case and 21 msle for the MIS11 configuration.

Additional experiments with prescribed present-day ice sheets or applying a freshwater hosing/extraction in the North Atlantic Ocean to alter AMOC strength are also conducted for selected orbital forcing–$CO_2$ configurations (Table 1).

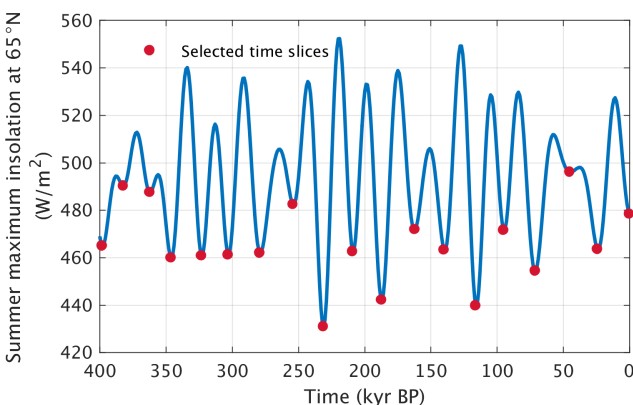

**Figure 1.** Summer maximum insolation at 65°N ($smx65$) in the period $400\,\mathrm{kyrBP}$ – present, computed using Laskar et al. (2004) data for Earth orbital parameters. Marked in red are all the 19 local minima of the time-series, whose orbital configurations are selected for the CLIMBER-X experiments.

## 3   Results

### 3.1   Critical $smx65$–$CO_2$ relationship for triggering glacial inception

The usage of $smx65$ as a proxy for orbital forcing is common practise and considered a good metric for the analysis of glacial inception processes (Ganopolski et al., 2016; Paillard, 1998; Leloup and Paillard, 2022), as the 65°N latitude coincides with the presence of land potentially capable of supporting the initial growth of ice sheets.

Considering $smx65$ as orbital forcing indicator, Fig. 2 shows the critical $smx65$–$CO_2$ combinations for triggering glacial inception in the 19 individual sets of experiments. Accounting for the fact that the radiative effect of $CO_2$ is logarithmic and that the temperature response to insolation is linear, we fit the data (via least-squares) with curves of the shape:

$$smx65_{\mathrm{cr}} = \alpha \ln \frac{CO_2}{280} + \beta. \tag{1}$$

The least-squares-fit of the data points produces values of $\alpha = -75\,\mathrm{W m}^{-2}$ and $\beta = 465\,\mathrm{W m}^{-2}$. The fit is good, with a coefficient of determination R$^2$=0.95. The largest error in the fit is produced for an intermediate insolation case ($462\,\mathrm{W m}^{-2}$, corresponding to the minimum in summer solstice insolation at 209 ka), in which the difference between the least-squares fit and the CLIMBER-X result for the critical insolation is $8\,\mathrm{W m}^{-2}$. This deviation from the best fit is explained by a very high obliquity at that time (see section 3.3). The new CLIMBER-X estimation for $\alpha$ and $\beta$ is very similar to the one generated with the CLIMBER-2 model and presented in Ganopolski et al. (2016). For that study the criterion for defining glacial inception was comparable (a growth of the NH ice sheets of more than 13.5 msle was required), and the values for $\alpha$ and $\beta$ were -77 and $466\,\mathrm{W m}^{-2}$, respectively. The robustness of the estimated parameters of the critical insolation–$CO_2$ relationship is also supported by the theoretical analysis presented in Appendix A, where it is shown that for climate conditions similar to

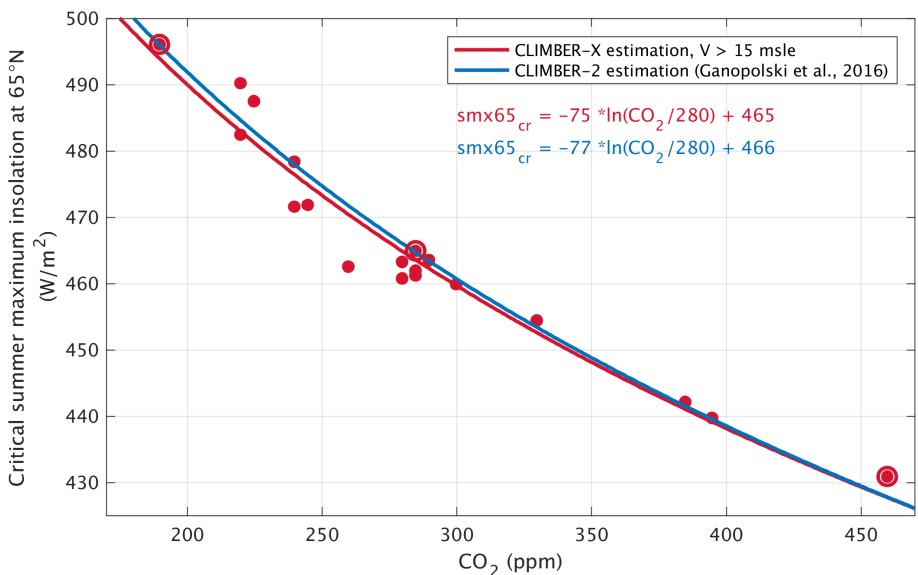

**Figure 2.** CLIMBER-X estimation of critical summer maximum insolation at 65°N ($smx65$) and $CO_2$ for triggering glacial inception at the threshold of 15 msle (red dots). The red line corresponds to the least-square fit following a logarithmic shape. The CLIMBER-2 estimation from Ganopolski et al. (2016) is depicted in blue. Big red circles indicate the experiments Hsmx65_LCO$_2$, Ismx65_ICO$_2$ and Lsmx65_HCO$_2$.

preindustrial, the value of $\alpha$ is about -80 $\mathrm{Wm^{-2}}$. Moreover, the value derived from the results presented by Abe-Ouchi et al. (2013) is -83 $\mathrm{Wm^{-2}}$. (Note that this number was not reported in Abe-Ouchi et al. (2013) but can be calculated from their Fig. 2). As was shown in Ganopolski et al. (2016), the value of $\beta$ is reasonably well constrained by paleoclimate data since the critical insolation curve must pass between rather close insolation values corresponding to the end of MIS11 and present insolation. At the same time, paleo data provide no constraint on the value of $\alpha$. This is why, in Talento and Ganopolski (2021), we used a very conservative approach by accepting as "valid" any $\alpha$ values from the range -150 to 0 $\mathrm{Wm^{-2}}$, i.e. we assumed relative uncertainties of up to 100%. The results of the present study strongly indicate that the uncertainty is much smaller, likely, not higher than 20%. Such a reduction of the uncertainty range would also significantly reduce the uncertainties in the projections of the timing of future glaciations for different anthropogenic $CO_2$ emissions.

In order to better understand glacial inception at the critical insolation–$CO_2$ threshold, and to evaluate similarities and differences for diverse insolation/$CO_2$ pairings, we select three experiments and analyse the temporal and spatial evolution of ice sheet growth. The three experiments (Hsmx65_LCO$_2$, Lsmx65_HCO$_2$ and Ismx65_ICO$_2$) correspond to either the highest, lowest or an intermediate value of the $smx65$ time-series 19 local minima, respectively. In each experiment, the atmospheric $CO_2$ corresponds to the critical $CO_2$ for triggering glacial inception (Table 1). While, by definition, all three experiments Hsmx65_LCO$_2$, Lsmx65_HCO$_2$ and Ismx65_ICO$_2$ reach at least 15 msle NH ice volume at some point during the simulation, they do it with different temporal and spatial dynamics (Fig. 3 and Fig. 4).

**Table 1.** Summary of selected experiments' configurations

| Experiment name | $smx65$ (Wm$^{-2}$) | $CO_2$ (ppm) | Ice sheets | Freshwater hosing/extraction into North Atlantic (Sv) |
|---|---|---|---|---|
| Control | 478.3 | 280 | Interactive | - |
| Control_FixIce | 478.3 | 280 | Fixed to present-day | - |
| Hsmx65_LCO$_2$ | 496.0 | 190 | Interactive | - |
| Hsmx65_LCO$_2$_FixIce | 496.0 | 190 | Fixed to present-day | - |
| Hsmx65_LCO$_2$+0.01Sv | 496.0 | 190 | Interactive | 0.01 |
| Hsmx65_LCO$_2$-0.01Sv | 496.0 | 190 | Interactive | -0.01 |
| Ismx65_ICO$_2$ | 464.8 | 285 | Interactive | - |
| Ismx65_ICO$_2$_FixIce | 464.8 | 285 | Fixed to present-day | - |
| Ismx65_ICO$_2$+0.01Sv | 464.8 | 285 | Interactive | 0.01 |
| Ismx65_ICO$_2$-0.01Sv | 464.8 | 285 | Interactive | -0.01 |
| Lsmx65_HCO$_2$ | 430.8 | 460 | Interactive | - |
| Lsmx65_HCO$_2$_FixIce | 430.8 | 460 | Fixed to present-day | - |
| Lsmx65_HCO$_2$+0.01Sv | 430.8 | 460 | Interactive | 0.01 |
| Lsmx65_HCO$_2$-0.01Sv | 430.8 | 460 | Interactive | -0.01 |

For the experiment Hsmx65_LCO$_2$ ($smx65 = 496$ Wm$^{-2}$ and $CO_2$=190 ppm) the NH ice volume time-series crosses the 15 msle threshold 67 kyrs into the simulation and reaches a quasi-equilibrium value of 16.6 msle only after 90 kyrs (Fig. 3). At the moment of crossing the glacial inception threshold (15 msle), the ice-sheets have significantly developed outside of Greenland over Iceland, Svalbard, Novaya Zemlya, and the Scandinavian Peninsula and, in North America, north of Hudson Bay, from 90°W to 70°W (Fig. 4a). The Scandinavian ice sheet development is aided by a weak AMOC (Fig. 5) and subsequent weakened meridional heat transport from the tropics towards that region. The AMOC reaches quasi-equilibrium conditions in the second half of the simulation, with a maximum magnitude of ∼16 Sv. At the threshold crossing time, the AMOC is weak and shallow (Fig. 6a) and deep convection in the North Atlantic Ocean only occurs in the Norwegian Sea (Fig. 6d).

In the experiment Ismx65_ICO$_2$ ($smx65 = 464.8$ Wm$^{-2}$ and $CO_2$=285 ppm) the NH ice volume time-series shows a faster increasing rate at the start of the experiment than the Hsmx65_LCO$_2$ experiment, but stagnates at 14.5 msle between 40 and 60 kyrs of simulation. At 60 kyrs of simulation, the time-series shows an abrupt increase, crossing the 15 msle threshold at time 63 kyrs. After that, the time-series stagnates again at 17.5 msle before showing another abrupt increase towards the end of the simulated period. In the 100 kyrs of simulation, quasi-stationary conditions are not reached (Fig. 3). In this experiment, the AMOC is unstable and oscillates between ∼25 Sv and ∼16Sv (Fig. 5). It should be noted that the timescale of these simulated oscillations is distorted by the climate acceleration factor. The AMOC weakening episodes precede the abrupt NH ice volume growths and are, thus, likely triggers. At the moment when the NH ice volume is 15 msle, the ice-sheets cover north of Hudson Bay from 100°W to 70°W (covering almost the totality of Baffin Island), full ice coverage of Iceland, Svalbard and Novaya

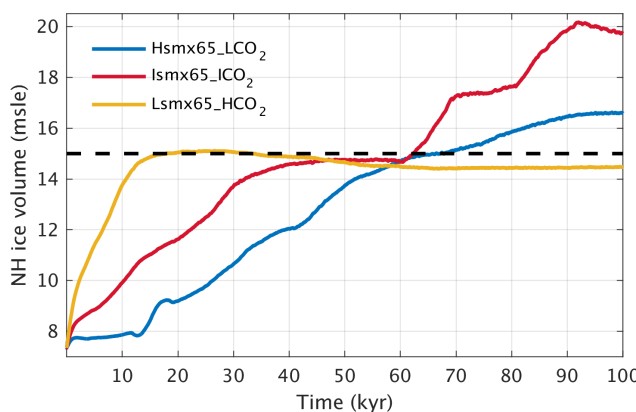

**Figure 3.** NH ice volume (msle) temporal evolution at the critical combination of $smx65$ and $CO_2$ for triggering glacial inception (threshold 15 msle level, dashed line), for three selected experiments (see Table 1).

Zemlya, and partial ice coverage of Scandinavia (Fig. 4c). The moderate ice growth over Scandinavia is related to the AMOC weakening episodes (Fig. 5) and the consequential decreased heat transport from the tropics, as such ice growths is inhibited in simulations with stable AMOC (see Section 3.2 with extraction simulation Ismx65_ICO$_2$-0.01Sv ). At the time of the 15 msle crossing, the Atlantic overturning circulation penetrates until $\sim$3000 m of depth and deep water formation does not occur in the Labrador Sea and south of Greenland (Fig.6b,e).

The Lsmx65_HCO$_2$ experiment ($smx65 = 430.8\,\mathrm{Wm}^{-2}$ and $CO_2$=460 ppm) shows the fastest NH ice volume time-series increase at the start of the simulation, crossing the 15 msle threshold after only 18 kyrs of simulation. After that, the ice volume slightly decreases, stabilising at 14.5 msle (Fig. 3). At the moment of crossing of the threshold, the ice sheet coverage over North America is larger than in the other two experiments, while no ice sheet develops over Scandinavia (Fig. 4c). For this case, the AMOC reaches equilibrium with a maximum strength of 27 Sv in the second half of the simulation (Fig. 5). Deep convection is widespread in the North Atlantic, occurring over Labrador Sea, south of Greenland and Norwegian Sea (Fig. 6f). The Atlantic overturning circulation reaches depths of 4000m (Fig. 6c).

From these experiments it is clear that the timescales involved with glacial development in the vicinity of the glacial inception bifurcation point are extremely long and that even 100 kyrs of simulation might not be enough to reach stationary conditions. It is also clear that, in CLIMBER-X, glacial inception can occur for different AMOC configurations. The spatial patterns of glaciation are different and related to the AMOC intensity, with a weak AMOC transporting less heat from the tropics towards Scandinavia and facilitating ice growth there. We see that in quasi-equilibrium conditions higher atmospheric $CO_2$ concentrations produce a stronger AMOC in CLIMBER-X. This was also observed in CLIMBER-2 (Bouttes et al., 2012; Gottschalk et al., 2019) and general circulation models, starting from Stouffer and Manabe (2003), who found AMOC strengthening under doubling and quadrupling of $CO_2$ and significant AMOC weakening under $CO_2$ halving in multimillennial simulations. Recently, Bonan et al. (2022) described the AMOC response to instantaneous $CO_2$ quadrupling in several state-of-the-art climate models. Mots of runs were not long enough, and AMOC continues to evolve through the runs, but at least in one model

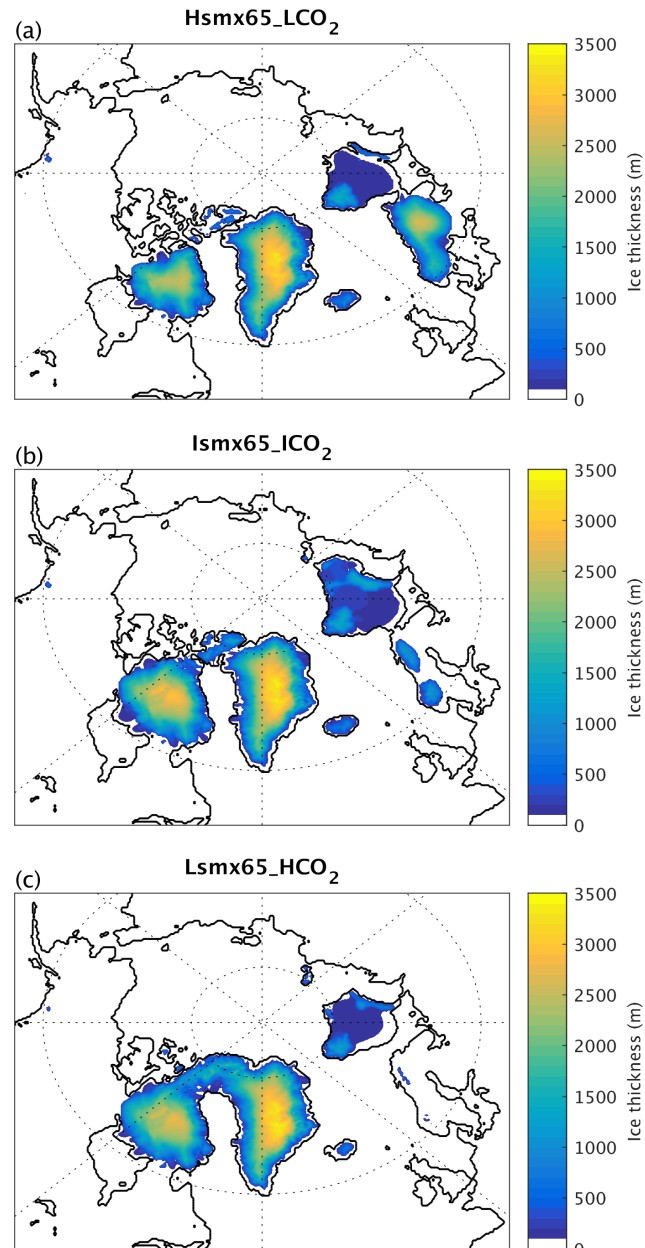

**Figure 4.** Ice thickness (a-c) for the Hsmx65_LCO$_2$, Ismx65_ICO$_2$ and Lsmx65_HCO$_2$ at the time when each experiment reaches the 15 msle threshold.

(CESM1), the AMOC is appreciably stronger under $CO_2$ quadrupling. For lower $CO_2$ concentration, there are two systematic studies (Oka et al., 2012; Galbraith and de Lavergne, 2019) where it has been shown that global cooling causes a weakening of the AMOC.

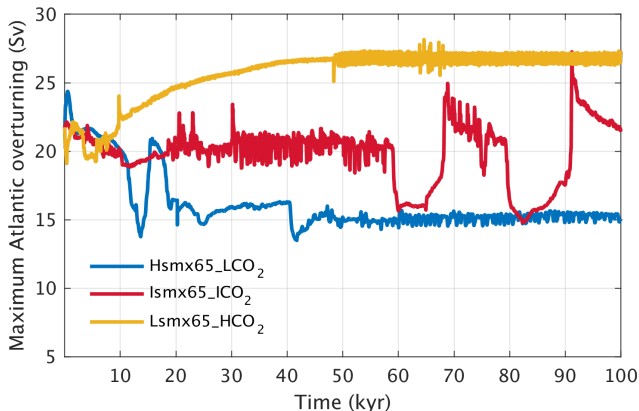

**Figure 5.** Maximum Atlantic Overturning (Sv) temporal evolution at the critical combination of $smx65$ and $CO_2$ for triggering glacial inception (threshold 15 msle level), for three selected experiments (see Table 1).

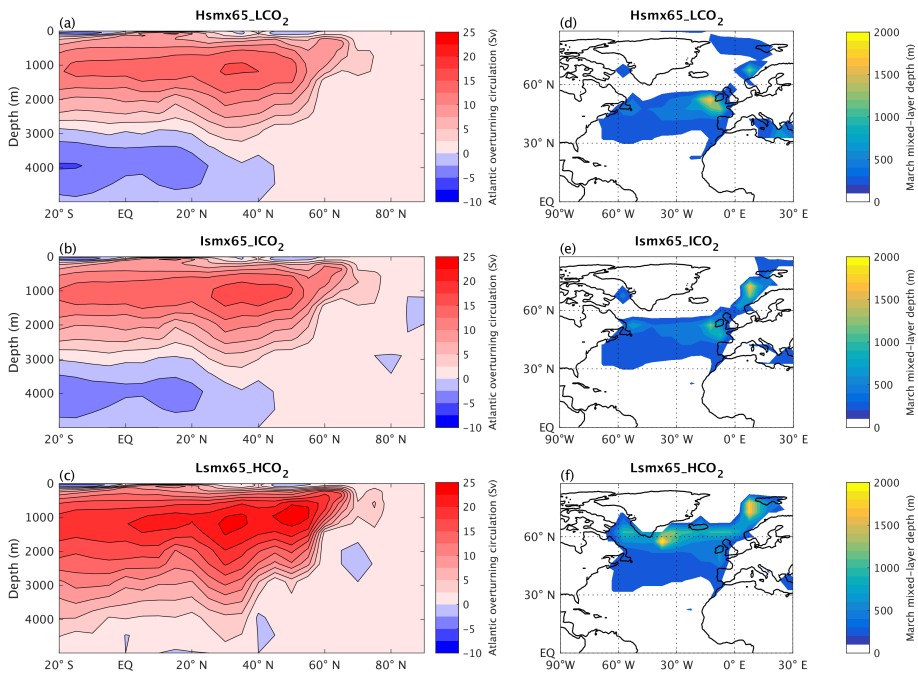

**Figure 6.** Annual mean Atlantic Ocean meridional overturning circulation streamfunction (a-c) and maximum (March) mixed layer depth (d-f) for the experiments Hsmx65_LCO$_2$, Ismx65_ICO$_2$ and Lsmx65_HCO$_2$ at the time when each experiment reaches the 15 msle threshold.

To get a better understanding on how different spatial patterns of ice growth arise with different combination of insolation and $CO_2$, we analyse the climatic conditions leading to the ice development. Figure 7 displays the annual-mean surface-mass balance (SMB) anomalies with respect to Control at the moment when the simulations reach the 15 msle of NH ice volume. For the three experiments, positive SMB anomalies with respect to Control are found over Greenland and over Baffin Island,

the anomalies being larger in magnitude for the Lsmx65_HCO$_2$ case. For the Hsmx65_LCO$_2$ experiment, the largest positive anomalies occur over the Scandinavian Peninsula.

Given the fact that surface temperature and precipitation fields might be directly impacted by the presence of an ice sheet, we generate auxiliary experiments in which the conditions are the same as in Hsmx65_LCO$_2$, Ismx65_ICO$_2$ and Lsmx65_HCO$_2$ but without interactive ice sheets (i.e. the ice sheets remain constant and equal to the initial present-day conditions; Table 1) and denote them with an additional FixIce label.

Boreal summer (June-August; JJA) 2m temperature and annual precipitation anomalies with respect to Control in simulations without interactive ice sheets for the NH between 120°W and 60°E are displayed in Fig. 8. The largest temperature differences between experiments are seen over the northern North Atlantic and are related to the differences in AMOC strength and related northward oceanic heat transport. The AMOC is the weakest in Hsmx65_LCO$_2$_FixIce, and as a result, Scandinavia is colder in this experiment by 2-4 °C than in the Control experiment. This explains why the ice sheets developed in Scandinavia in Hsmx65_LCO$_2$ but not in Lsmx65_HCO$_2$. Over northern North America, temperatures are rather similar in all experiments, which is explained by the fact that summer temperatures are the main factor controlling the mass balance of ice sheets. Still, temperatures in Hsmx65_LCO$_2$ lower by ca. 2 °C than in Lsmx65_HCO$_2$_FixIce. The reason why glacial inception requires lower temperatures in experiment with higher insolation is straightforward. Unlike CO$_2$, insolation affects the surface mass balance of ice sheets not only through temperature but also directly through absorbed shortwave radiation (e.g. Robinson et al., 2010). This is why, to compensate the effect of higher insolation on surface ice sheet melt in the experiment Hsmx65_LCO$_2$, summer temperatures should be lower than in the experiments where glacial inception is simulated under lower summer insolation.

Annual precipitation anomalies relative to control shown in Fig. 8d-f also mostly reflect changes in AMOC, although globally there is more precipitation under a higher CO$_2$ level. Over Scandinavia, precipitation changes counteract temperature changes, but the effect of temperature is dominant. This is why, in spite of significantly lower annual precipitation, the ice sheet in Scandinavia grows in the experiment Hsmx65_LCO$_2$ but not in Lsmx65_HCO$_2$. Precipitation differences in northern North America are small.

In general, a comparison of these three experiments shows that summer climate conditions corresponding to the initiation of glaciation are similar in North America despite very different insolation and CO$_2$ in these experiments. This explains rather similar patterns of initial ice sheet growth in North America. Climate conditions are more different in Scandinavia, and this is why glaciation develops here simultaneously with glaciation in North America only under sufficiently low CO$_2$.

## 3.2 Role of AMOC change in glacial inception

As shown above, the AMOC state is different for different combinations of insolation–CO$_2$ close to the glacial inception limit (Fig. 6a-c). The question therefore arises on what role the AMOC plays for glacial inception. To explore the relative importance of the AMOC state compared to the summer insolation and the atmospheric CO$_2$ concentration in shaping the conditions for glacial inception, we have performed additional simulations in order to separate the contribution of the different factors. For

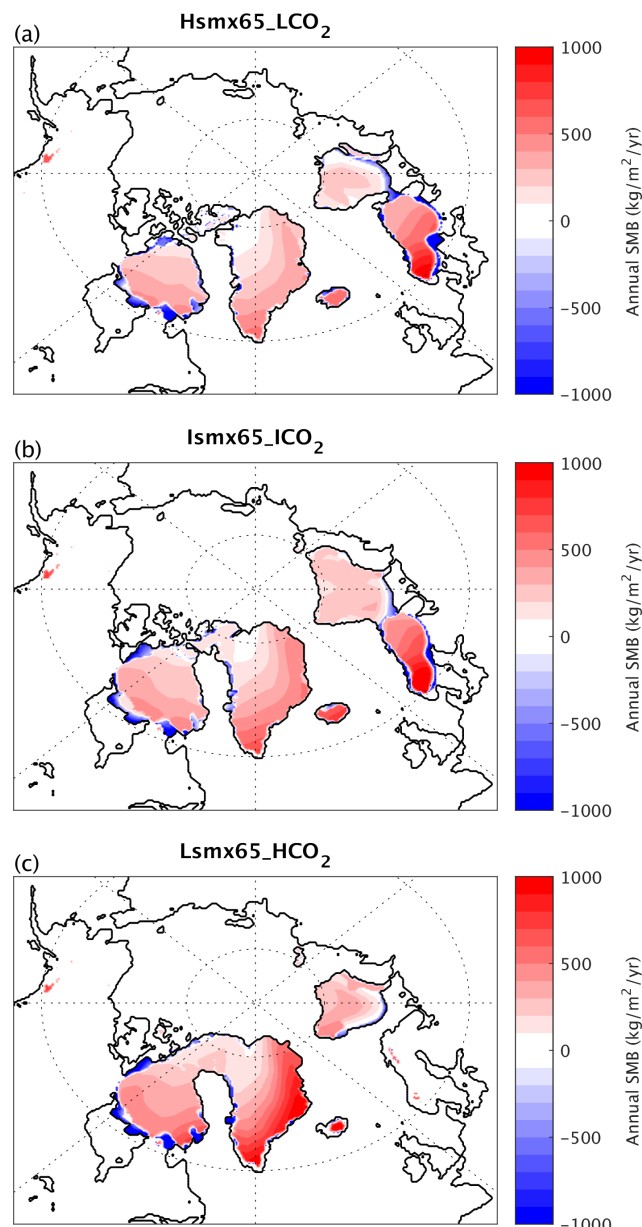

**Figure 7.** Annual surface mass balance anomaly with respect to Control for the experiments Hsmx65_LCO$_2$, Ismx65_ICO$_2$ and Lsmx65_HCO$_2$ (a-c), respectively, at the time when each experiment reaches the 15 msle threshold. Continental and ice sheet margins are show in black.

that we have taken the extreme cases, Hsmx65_LCO$_2$ and Lsmx65_HCO$_2$, and run three sets of model simulations with prescribed present-day ice sheets:

- present-day orbital configuration; CO$_2$ concentration corresponding to HCO$_2$ and LCO$_2$,

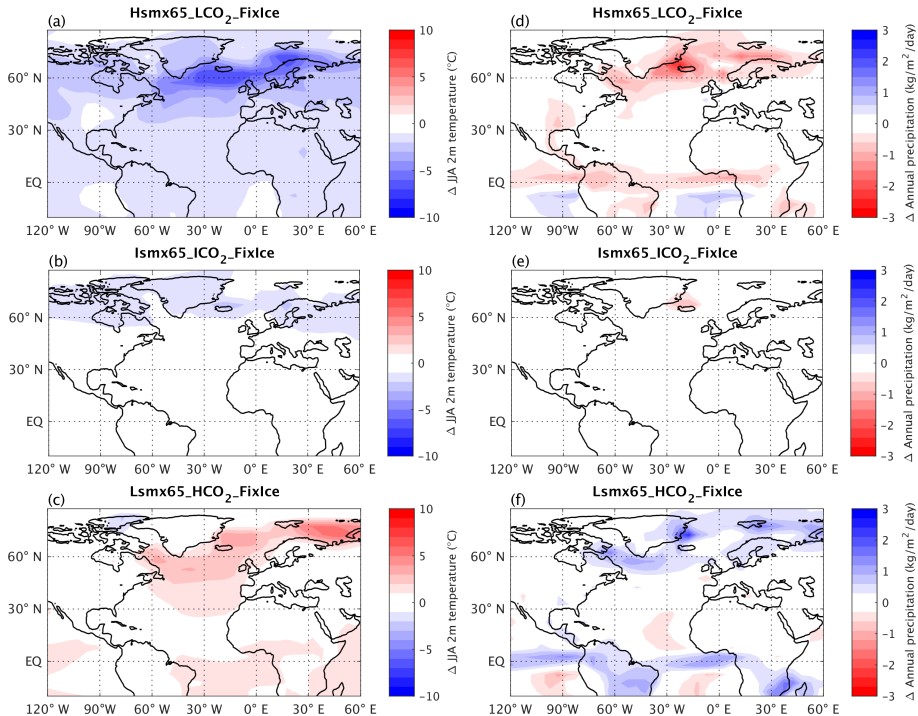

**Figure 8.** Summer (JJA) temperature anomalies (a-c) and annual precipitation anomalies (d-f) with respect to Control_FixIce for experiments Hsmx65_LCO$_2$_FixIce, Ismx65_ICO$_2$_FixIce and Lsmx65_CO$_2$_FixIce. Control continental margins (present-day conditions) are shown in black.

– pre-industrial CO$_2$ concentration; orbital configuration corresponding to Hsmx65 and Lsmx65,

     – pre-industrial CO$_2$ concentration and orbital configuration; with and without 0.1 Sv freshwater hosing in the North Atlantic between 50–70°N to get two states with weak and strong AMOC ($\sim$10 Sv difference, roughly corresponding to the difference between Hsmx65_LCO$_2$ and Lsmx65_HCO$_2$ experiments) under the same background climate conditions.

These three sets of two simulations each allow to separate the effect of atmopheric CO$_2$, summer insolation and AMOC state
on the summer temperature in the NH. The results clearly show that summer temperatures at the locations where the ice sheets start to growth during glacial inceptions (i.e. Scandinavia and northeastern Canada), are much more sensitive to CO$_2$ and orbital forcing than to changes in the AMOC (Fig. 9). The AMOC state has a generally larger impact on summer temperatures over Scandinavia (Fig. 9c), at least partly explaining why the inception patterns show largest differences there (Fig. 4). It has to be noted that changes in CO$_2$ and insolation also affect the AMOC. In particular, the AMOC is stronger by 9 Sv in the
experiment with high CO$_2$ compared to low CO$_2$ and by 3 Sv in experiments with high insolation compared to low insolation. However, as seen from Fig. 9c, these changes can contribute only a little to the direct effect of CO$_2$ (Fig. 9a) and insolation (Fig. 9b). Thus, at the potential locations of glacial inception, AMOC changes provides a positive, but not very strong, feedback to both primary forcings.

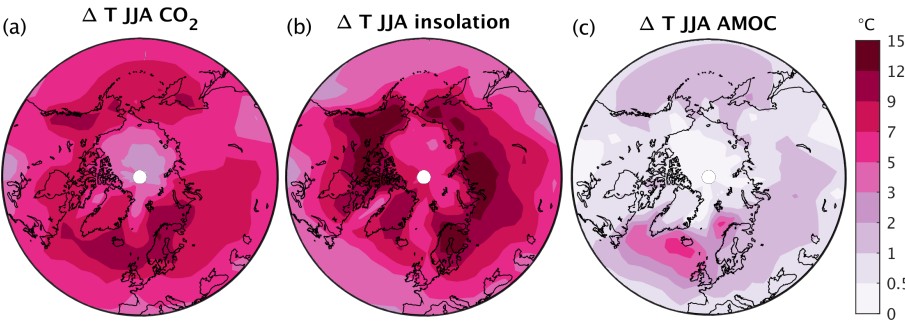

**Figure 9.** Summer (JJA) temperature differences due to (a) $CO_2$ ($HCO_2$-$LCO_2$), (b) insolation (Hsmx65-Lsmx65) and (c) AMOC state (strong-weak).

To further investigate the role of AMOC strength change at the glacial inception bifurcation limits, we produce a series of water hosing/extraction experiments (Table 1). The freshwater forcing is applied in the North Atlantic Ocean between 50°N and 70°N and compensated over the tropical (30°S-30°N) Pacific Ocean to prevent a global salinity drift. The magnitude of the freshwater flux is $0.01\,\mathrm{Sv}$, either positive (added freshwater, leading to a weakening of AMOC) or negative (extracted freshwater, leading to a strengthening of AMOC). The hosing/extraction forcing is applied for the whole duration of the experiments. Note, that $0.01\,\mathrm{Sv}$ is a very small value compared to a typical stability threshold for AMOC in the CLIMBER-X model (Willeit et al., 2022).

The freshwater extraction produces less NH ice sheet volume in all three cases (Fig. 10). In particular, in the experiments Hsmx65_$LCO_2$ and Ismx65_$ICO_2$ the extraction in enough to prevent the NH ice volume to cross the $15\,\mathrm{msle}$ threshold used for the definition of glacial inception. In opposition, the freshwater hosing flux produces in all three cases a faster and larger ice growth in the NH. The largest change in terms if NH ice volume is produced for the Ismx65_$ICO_2$, reaching a magnitude of $\sim150\,\mathrm{msle}$. In this case, the freshwater hosing flux produces the largest AMOC weakening of all three experiments, and even an almost complete AMOC shutdown between 50 and $80\,\mathrm{kyrs}$ of the simulation (Fig. 11).

This set of hosing experiments shows that AMOC (in particular its stability) plays a role in the triggering of glacial inception and the rate of ice growth. In general, the weaker the AMOC the larger and faster the development of the NH ice sheets, for a given insolation/$CO_2$ pairing.

## 3.3 Suitability of $smx65$ as proxy for orbital forcing

The usage of $smx65$ as a proxy for orbital forcing in tracing the glacial inception bifurcation is justified by the relevance of the 65°N latitude for the development of ice sheets (Ganopolski, 2024). However, as any simple metric for such a complex phenomenon as the influence of orbital forcing on Earth climate, it cannot be perfect. This is why it is not surprising that although most of the points in Fig. 2 are located very close to the proposed logarithmic curves, several deviate more significantly, for example, the point corresponding to insolation at $209\,\mathrm{ka}$, as it was discussed above.

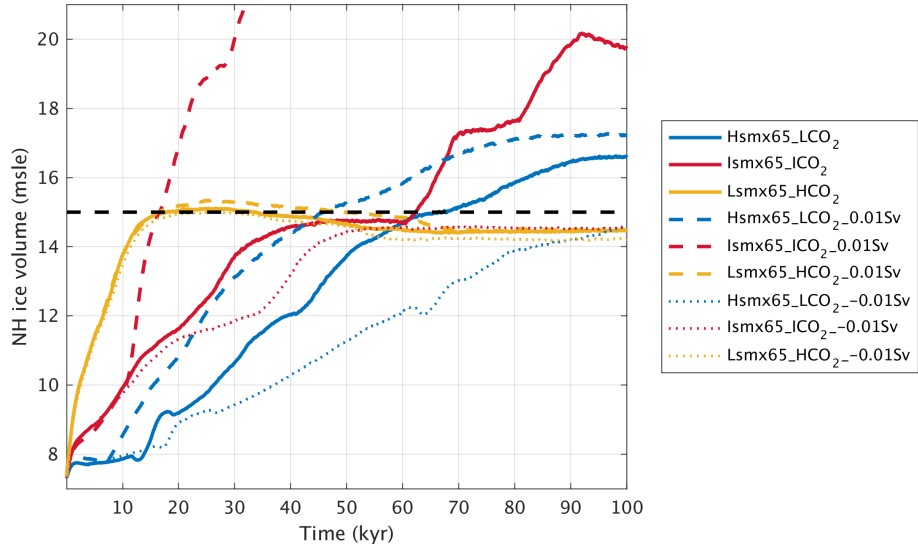

**Figure 10.** NH ice volume (msle) temporal evolution in water hosing/extraction experiments (Table 1).

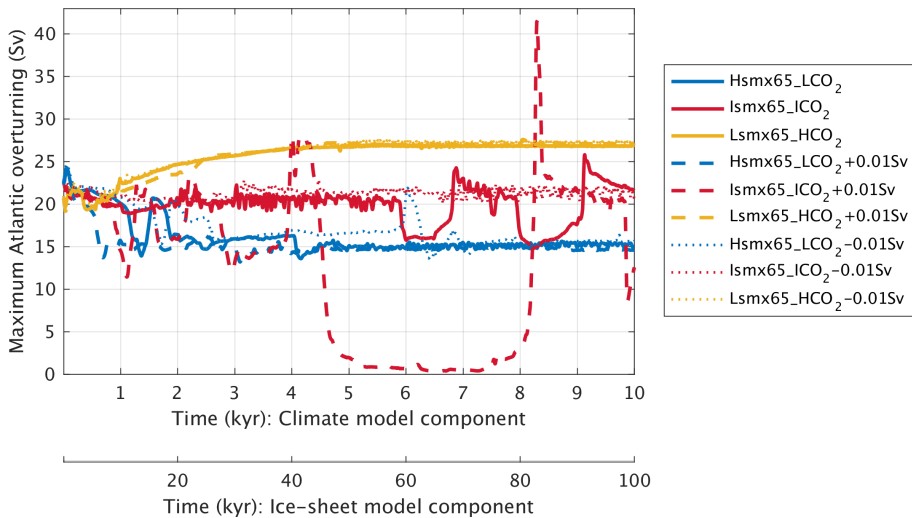

**Figure 11.** Maximum Atlantic overturning (Sv) temporal evolution in water hosing/extraction experiments (Table 1).

To evaluate the possibility of improving the proxy for orbital forcing, we separate the $smx65$ time-series into its obliquity and precessional components (as in Jackson and Broccoli (2003)). The decomposition is done via a least-squares fit to the data of the last $400\,\mathrm{kyrs}$, expressing the $smx65$ time-series in the form:

$$smx65 = a_0 + a_1 \cdot ecc \cdot \sin\omega + a_2 \cdot (obl - \overline{obl}), \tag{2}$$

with ecc, $\omega$ and obl denoting eccentricity, longitude of perihelion and obliquity, respectively; coefficients $a_0$, $a_1$, and $a_2$ are derived from the least-squares fit and the over bar indicates time averaging. The precessional ($smx65_p$) and obliquity ($smx65_o$) components are defined as:

$$smx65_p = a_0 + a_1 \cdot ecc \cdot \sin\omega. \tag{3}$$

$$smx65_o = a_2 \cdot (obl - \overline{obl}). \tag{4}$$

We analyse different linear combinations of $smx65_p$ and $smx65_o$ as alternative proxies for orbital forcing. We wish to find $\gamma$, which maximises the goodness-of-fit ($R^2$-coefficient of determination) between the logarithm of critical $CO_2$ and the generalized proxy for summer insolation in the form: $smx65_p + \gamma smx65_o$.

Any value of $\gamma$ between 1 and 2 yields an extremely good fit ($R^2$ higher than 0.95) and, thus, the quantities $smx65_p + \gamma smx65_o$ for those values of $\gamma$ are good proxies for orbital forcing (Fig. 12). Values of $\gamma$ lower than 1, in which the contribution of obliquity is considered relatively less important than that of precession, are not optimal proxies. Similarly, values of $\gamma$ higher than 2 (i.e. the obliquity contribution is weighted more than the precessional one) are also not adequate. The best fit is found for $\gamma \approx 1.3$. Although this fit is only marginally better than for $\gamma = 1$, corresponding to $smx65$, and thus of little importance for the main results presented in this paper, it still can help to explain some deviations of individual results from the logarithmic curve. Indeed, as was discussed in Ganopolski (2024), the $smx65$ metric contains the smallest contribution of obliquity compared to other proposed proxies for orbital forcing. The results of the analysis presented above suggest that the relative contribution of obliquity is underestimated in $smx65$ by ca. 30%. Since the amplitude of obliquity contribution to $smx65$ is only about $15\,\mathrm{Wm}^{-2}$, the generalized proxy for summer insolation for $\gamma = 1.3$ would not deviate from $smx65$ by more than $5\,\mathrm{Wm}^{-2}$, which is very small compared to $smx65$ values. However, it has an implication for the "outlier" in Fig. 2 corresponding to insolation at 209 ka, when obliquity was close to its maximum value during the entire past 800 ka. The correction upwards by $5\,\mathrm{Wm}^{-2}$ would bring the point corresponding to 209 ka insolation much closer to the logarithmic curve.

## 4 Summary and conclusions

We used the recently developed EMIC CLIMBER-X to produce a new estimation of the critical insolation – $CO_2$ relationship for triggering glacial inception. The use of CLIMBER-X for tracing the glacial inception bifurcation constitutes a further development from previous attempts produced with the coarser-resolution and simpler CLIMBER-2 model (Ganopolski et al., 2016; Archer and Ganopolski, 2005).

The study is based on sets of equilibrium experiments in which the orbital parameters and atmospheric $CO_2$ concentration are kept constant in time. The sets are based on orbital configurations corresponding to all the local minima of summer maximum insolation at 65°N ($smx65$) of the last 400 kyrs. Therefore, our experiments reflect a diverse arrangement of orbital parameters in configurations potentially favourable for glacial inception. The estimation of the glacial inception bifurcation is done within a 5 ppm resolution in the $CO_2$ phase-space.

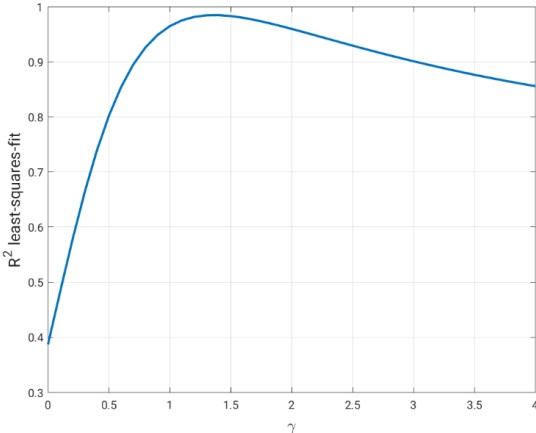

**Figure 12.** Goodness-of-fit ($R^2$ coefficient of determination) derived from the least-squares-fit between $\ln(CO_{2,\mathrm{critical}})$ and $smx65_p + \gamma smx65_o$ as a function of $\gamma$, considering $CO_2$ values critical for glacial inception at the $15\,\mathrm{msle}$ level.

The new estimation of the critical insolation – $CO_2$ relationship for triggering glacial inception follows a logarithmic dependence, explained by the logarithmic shape of the radiative forcing of $CO_2$. When considering that glacial inception occurs if the NH ice volume develops at least $15\,\mathrm{msle}$, the new CLIMBER-X estimation for the critical summer maximum insolation
at $65°$N and $CO_2$ relationship is:

$$smx65_{\mathrm{cr}} = -75 \cdot \ln \frac{CO_2}{280} + 465. \tag{5}$$

This new estimation is close to the one produced with CLIMBER-2 in Ganopolski et al. (2016), who used a similar threshold for NH ice volume. Moreover, we showed that the summer maximum insolation at $65°$N is a skilful single metric for tracing the glacial inception bifurcation.
We also showed that the temporal and spatial patterns of glacial inception depend on the combination of insolation and critical $CO_2$ concentration. In all simulations of glacial inception, ice sheets developed north of Hudson Bay in North America, in Iceland and Svalbard. Considerable ice growth over the Scandinavian Peninsula is only observed in experiments in which the Atlantic Meridional Overturning Circulation (AMOC) significantly weakens, which occurs in either low $CO_2$ cases (stable weak AMOC) or intermediate $CO_2$ situations (unstable AMOC with substantial weakening episodes). In general, the lower
the $CO_2$, the weaker the AMOC and, thus, the weaker the heat transport from the tropics towards European high latitudes, facilitating the Scandinavian ice sheet development. The time needed for reaching the glacial inception bifurcation threshold varies between 18 and almost 100 thousand years, highlighting the long timescales of the climate-cryosphere system at the bifurcation limit.

*Code and data availability.* Code and data used in this study can be obtained from https://doi.org/10.17605/OSF.IO/2MHKY (Talento, 2023).

## Appendix A: Theoretical derivation of the critical insolation–$CO_2$ relationship

To derive the critical insolation-$CO_2$ relationship for triggering glacial inception, we consider the surface energy balance of snow cover at the location where the nucleation of the northern continental ice sheet begins during glacial inceptions. Important assumptions are:

– this location remains the same for the different combinations of insolation and $CO_2$,

– glacial inception begins from the formation of extensive perennial snowfields rather than from the spreading of glaciers from high mountains.

Both these assumptions are consistent with the results of our model simulations. Surface daily mass balance of the snowpack during the melt season is described by the following equation:

$$S_{\mathrm{abs}} + R_{\downarrow} - R_{\uparrow} + H = ML, \tag{A1}$$

where $S_{\mathrm{abs}}$ is the absorbed short-wave radiation, $R_{\downarrow}$ is the downward long-wave radiation flux near the earth surface, $R_{\uparrow}$ is the longwave radiation emitted by the snow surface, $H$ is the sensible heat flux (positive downward), $M$ is the snow melt rate in $\mathrm{kg\,m^{-2}s^{-1}}$, and $L$ is the latent heat of fusion. Surface latent heat flux can be neglected during the melt season, as it is an order of magnitude smaller than short-wave and long-wave radiation fluxes (e.g. Ettema et al., 2010). Since we consider only the period of snow-melt, surface temperature during this period is $0\,^{\circ}\mathrm{C}$. In this case, the terms in the left-hand side of the eq. (A1) can be expressed as follows:

$$S_{\mathrm{abs}} = S(1 - \alpha_{\mathrm{s}})\tau, \tag{A2}$$

$$R_{\downarrow} = F(0) + \beta T, \tag{A3}$$

$$R_{\uparrow} = \epsilon \sigma T_0^4, \tag{A4}$$

$$H = \gamma T, \tag{A5}$$

where $T$ is the surface air temperature (in $^{\circ}\mathrm{C}$), $S$ is the insolation at the top of the atmosphere, $\alpha_{\mathrm{s}}$ is the surface albedo, $\tau$ is the atmospheric transmissivity, $F(0)$ is the downward long-wave radiation for a surface air temperature of $0°\mathrm{C}$ in $\mathrm{W\,m^{-2}}$,

$T_0 = 273.15\,\mathrm{K}$, $\sigma = 5.67 \times 10^{-8}\,\mathrm{Wm^{-2}K^{-4}}$ is the Stefan-Boltzmann constant, $\epsilon \approx 1$ is the snow emissivity for long-wave radiation, $\alpha$ and $\beta$ are empirical parameters. In the nucleation centre of glacial inception (northern Canada and Baffin Island), the melt-season is only about two months and approximately coincides with the period of positive air temperatures. This allows us to make further simplifications when integrating eq. (A1) over the melt season after substituting eqs. (A2),(A3),(A4),(A5):

$$\overline{S}(1 - \alpha_{\mathrm{s}})\tau + F(0) + \beta\overline{T} - \sigma T_0^4 + \gamma\overline{T} = \overline{M}L, \tag{A6}$$

where $\overline{T}$ is the surface air temperature, $\overline{S}$ is daily insolation and $\overline{M}$ is snowmelt, all averaged over the melt season. Assuming that the duration of the melt season at the location of glacial inception is about two months (which is similar to present climate conditions in the area where glacial inception is simulated in CLIMBER-X), the seasonal surface air temperature evolution can be approximated by a sinusoidal function with zero air temperatures at the beginning and end of the melt season and the

365 average insolation and temperature can be expressed through their maximum annual values as:

$$\overline{S} = k_{\mathrm{S}} smx65, \tag{A7}$$

$$\overline{T} = k_{\mathrm{T}} T_{\max}, \tag{A8}$$

where $k_{\mathrm{S}} \approx 0.9$ and $k_{\mathrm{T}} \approx 0.7$ (see Fig. A1). Maximum summer air temperature is parameterized as

$$T_{\max} = T_{\max}^* + \delta \ln \frac{CO_2}{CO_{2_0}} + \mu \cdot (smx65 - smx65^*), \tag{A9}$$

where $T_{\max}^*$ is the maximum summer temperature at the location of glacial inception for preindustrial conditions, namely for $CO_2 = CO_{2_0}$ and $smx65 = smx65^*$, where $CO_2$ is atmospheric $CO_2$ concentration, $CO_{2_0} = 280\,\mathrm{ppm}$, and $smx65^* = 479\,\mathrm{Wm^{-2}}$ is the present-day maximum insolation at 65°N. The parameter $\mu$ represents the sensitivity of summer temperature at the location of glacial inception to summer insolation, i.e.:

$$\mu = \frac{\Delta T_{\max}}{\Delta smx65}. \tag{A10}$$

The parameter $\delta$ in eq. (A9) represents the sensitivity of maximum summer temperature to $CO_2$ concentration. It can be expressed through other known parameters as $\delta = p_{\mathrm{a}} r_{\mathrm{f}} c_{\mathrm{s}}$, where $p_{\mathrm{a}}$ is the "polar amplification" of maximum summer temperature equal to $\Delta T_{\max}/\Delta T_{\mathrm{g}}$, where $\Delta T_{\mathrm{g}}$ is the global air temperature anomaly caused by $CO_2$ concentration change, $r_{\mathrm{f}}$ is the parameter determining radiative forcing of $CO_2$:

$$\Delta R_{CO_2} = r_{\mathrm{f}} \ln \left( \frac{CO_2}{CO_{2_0}} \right) \tag{A11}$$

and $c_{\mathrm{s}}$ is the specific climate sensitivity (e.g. Rohling et al., 2012) which relates the global temperature change to radiative forcing:

$$\Delta T_{\mathrm{g}} = c_{\mathrm{s}} \Delta R. \tag{A12}$$

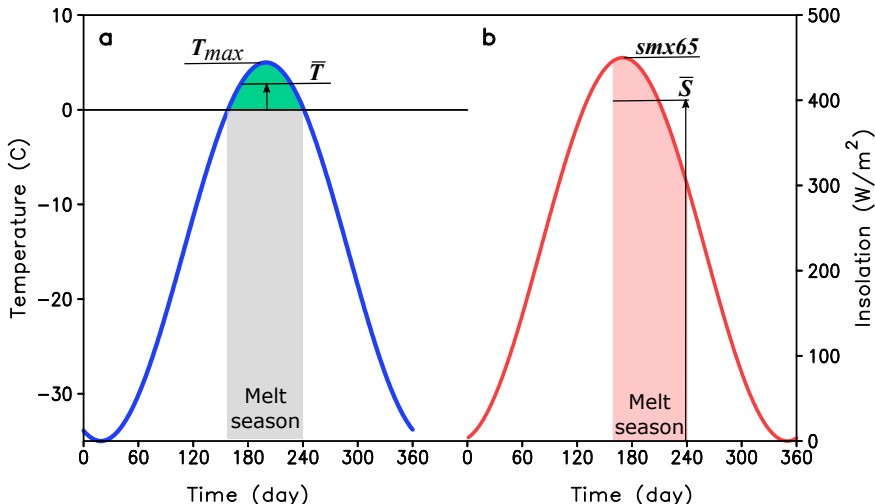

**Figure A1.** Typical seasonal variations of surface air temperature (a), and insolation (b) at the location of glacial inception.

Based on IPCC 2021 (2021) and climate model simulations $p_a = 1.5$, $r_f = 5.7\,\mathrm{Wm^{-2}}$, and $c_s = 0.75\,\mathrm{°CW^{-1}m^2}$, which is
equivalent to an equilibrium climate sensitivity of $3\,°C$. Thus $\delta = 6.4\,°C$. To derive a critical relationship between insolation
and $CO_2$ for triggering glacial inception, we make use of the fact that the glacial inception occurs when the total snowmelt
minus refreezing during the melt season does not exceed annual snowfall $P$ (in $\mathrm{kgm^{-2}}$), which for the considered region is
less than $200\,\mathrm{kgm^{-2}}$:

$$\overline{M}\Delta t_\mathrm{M} = \frac{P}{1 - f_\mathrm{r}}, \tag{A13}$$

where $\Delta t_\mathrm{M}$ is the duration of the melt season and $f_\mathrm{r}$ is an average refreezing fraction of snowmelt. Thus, eq. (A6) can be
re-written as:

$$k_\mathrm{S}(1-\alpha_\mathrm{s})\tau \cdot smx65 + k_\mathrm{T}(\beta+\gamma)\left(\delta\ln\frac{CO_2}{CO_{2_0}} + \mu \cdot smx65\right) = \frac{PL}{(1-f_\mathrm{r})\Delta t_\mathrm{M}} + \sigma T_0^4 - F(0) - k_\mathrm{T}(\beta+\gamma)\left(T^*_\mathrm{max} - \mu \cdot smx65^*\right),$$

$$\tag{A14}$$

which can be rewritten in the form:

$$smx65_\mathrm{cr} = A - B\ln\frac{CO_2}{CO_{2_0}}, \tag{A15}$$

which is identical to the relationship proposed in Ganopolski et al. (2016). Here

$$A = \frac{PL(1-f_\mathrm{r})^{-1}\Delta t_\mathrm{M}^{-1} + \sigma T_0^4 - F(0) + k_\mathrm{T}(\beta+\gamma)(\mu \cdot smx65^* - T^*_\mathrm{max})}{k_\mathrm{S}(1-\alpha_\mathrm{s})\tau + k_\mathrm{T}\mu(\beta+\gamma)} \tag{A16}$$

and

$$B = \frac{k_\mathrm{T}\delta(\beta+\gamma)}{(k_\mathrm{S}(1-\alpha_\mathrm{s})\tau + k_\mathrm{T}\mu(\beta+\gamma)}. \tag{A17}$$

We will start from determining $B$, which can be calculated using the following values of model parameters: $\alpha_s = 0.7$, which is a typical albedo of melting snow, $\tau = 0.5$ is taken from Robinson et al. (2010), $\beta = 4\,\mathrm{Wm^{-2\,\circ}C^{-1}}$ (derived from the regression of model data for the considered region), $\gamma = 15\,\mathrm{Wm^{-2\,\circ}C^{-1}}$ (middle of the range from Braithwaite (2009), $\mu = 0.08\,\mathrm{W^{-1}m^2\,\circ C}$ (from climate model simulations for different orbital parameters). However, since $k_S(1 - \alpha_s)\tau = 0.13 \ll k_T(\beta + \gamma) = 0.9$, eq. (A17) for $B$ can be significantly simplified, and only two parameters are required:

$$B = \frac{\delta}{\mu} \approx 80\,Wm^{-2}. \tag{A18}$$

Equation (A16) for $A$ contains a number of parameters which are not accurately known. However, knowledge of all these parameters is not required. It is only essential that the value of A for different glacial inceptions should remain approximately constant. This requirement is satisfied since only the total snowfall depends on the values of insolation and $CO_2$ in this formula. However, the contribution of the latent heat term $PL(1 - f_r)\Delta t_M$ to the value of $A$ is only about 10-20%, and, according to climate model simulations, changes in annual precipitation for a wide range of $CO_2$ concentrations are of the same magnitude. Thus, the total variations of $A$ across the considered range of climate conditions should not exceed few percents. This allows us to determine $A$ using two paleoclimate constraints, namely, that glacial inceptions occurred at the end of MIS 19 and 11, while no glacial inception occurred in the recent past (see also Fig. 3b in Ganopolski et al. (2016). These empirical constraints suggest that under the present-day maximum summer insolation of $479\,\mathrm{Wm^{-2}}$, glacial inception can only occur if $CO_2$ concentration is around $240\,\mathrm{ppm}$. This gives the following value:

$$A = 479 + 80\ln\frac{240}{280} \approx 467\,Wm^{-2}. \tag{A19}$$

Note that the choice of critical $CO_2$ for present day insolation of $240\pm20\,\mathrm{ppm}$, results in the uncertainty of A only $\pm6\,\mathrm{Wm^{-2}}$. Equation (A15) now can be rewritten as:

$$smx65_{cr}[Wm^{-2}] = -80 \cdot \ln\frac{CO_2[ppm]}{280} + 467, \tag{A20}$$

which is numerically very similar to Ganopolski et al. (2016) and the results of the present study.

Equation (A18) for $B$ has a clear physical sense: the slope of the critical relationship between insolation and the logarithm of $CO_2$ concentration is equal to the ratio between local temperature sensitivities to $CO_2$ ($\delta$) and summer insolation ($\mu$). This value has dimension $\mathrm{Wm^{-2}}$. If one wants to compare the effect of insolation to the effect of radiative forcing of $CO_2$ in the same units of $\mathrm{Wm^{-2}}$, then it would be $B/r_f = 15$, i.e. one $\mathrm{Wm^{-2}}$ of $CO_2$ forcing is approximately equivalent to $15\,\mathrm{Wm^{-2}}$ of the maximum summer insolation in determining the glacial inception. This number is consistent with the results of earlier studies (Calov and Ganopolski, 2005; Abe-Ouchi et al., 2013).

*Author contributions.* ST performed the model simulations and made the figures, with contributions from MW. All authors contributed to the analysis of results and preparation of the manuscript.

*Competing interests.* The authors declare that they have no conflict of interest.

*Acknowledgements.* Financial support for ST was provided by the Swiss National Cooperative for the Disposal of Radioactive Waste (NA-
430 GRA). MW is funded by the German climate modeling project PalMod supported by the German Federal Ministry of Education and Research (BMBF) as a Research for Sustainability initiative (FONA) (grant nos. 01LP1920B, 01LP1917D, 01LP2305B). The authors gratefully acknowledge the European Regional Development Fund (ERDF), the German Federal Ministry of Education and Research and the Land Brandenburg for supporting this project by providing resources on the high performance computer system at the Potsdam Institute for Climate Impact Research.

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
