# Peer review of "New estimation of critical insolation – $CO_2$ relationship for triggering glacial inception"

_Climate of the Past, 2023_

## Author Response (AR1)

**Referee#1**

***General comments***

*- From l. 175 to 237, the authors selected three combinations that cover the spectrum in terms of insolation / CO2 to achieve inceptions and they discuss the temporal and spatial evolutions of the ice*

*sheets. What we mostly see is that the AMOC is very different for these three simulations. Although I appreciate this section, I found it hard to compare directly these simulations because of the differences in terms of AMOC. I have the feeling that we are not really looking at the impact of insolation/CO2 on ice sheet evolution here but instead we are looking at the impact of inso/CO2 on the AMOC (and ultimately yes, the ice sheet evolution). Perhaps what could have been done is alternative experiments using larger*

*freshwater flux than the ones in Sec. 3.2, in order to have a shutdown state for these three simulations. In doing so, we would have quantified the impact of insolation/CO2 disregarding the state of the AMOC.*

We appreciate this reviewer's comment and agree that the role of individual factors should be discussed. To this end we performed a set of additional experiments with prescribed present-day ice sheets. Namely, we performed three sets of quasi-equilibrium experiments where 1) only orbital forcing was set to Hsmx65 and Lsmx65 values with constant 280 ppm $CO_2$ concentration; 2) $CO_2$ was set to LCO2 and HCO2 values with the present-day orbital forcing; 3) Water hosing experiment under present-day boundary conditions in which AMOC was weakening by ca. 10 Sv which corresponds to the difference between Hsmx65_LCO2 and Lsmx65_HCO2 experiments. Weaker AMOC state was obtained by applying of a constant freshwater hosing of 0.1 Sv in the northern North Atlantic. The results are shown in the figure below. The main conclusion is that summer temperatures at the locations where the ice sheets start to growth during glacial inceptions (i.e. Scandinavia and northeastern Canada) , are much more sensitive to CO2 and orbital forcing than to changes in the AMOC. It has to be noted that changes in $CO_2$ and insolation also affect AMOC. In particular, AMOC is stronger by 9 Sv in experiments with high $CO_2$ compared to low CO2 and by 3 Sv in experiments with high insolation compared to low insolation.

However, as it is seen from the right panel, these changes can contribute only a little to the direct effect of $CO_2$ (left panel) and insolation (middle panel). Thus, in both potential locations of glacial inception, AMOC changes provides positive, but not very strong, feedback to both primary forcings. We have added this discussion and the figure below in Sec. 3.2 of the revised paper.

[Figure]

**Fig. 1**. Summer temperature difference between experiments with high (460 ppm) and low (190 pp) $CO_2$ concentration (left panel); high (496 W/m$^2$) and low (430.8 W/m$^2$) maximum summer insolation at 65°N (middle panel ); two states in which AMOC strength differs by 10 Sv (right panel).

*- Sensitivity of the AMOC to CO2. The simulations presented here show a general weaker AMOC for lower CO2. The authors suggest that it was also the case in CLIMBER-2 and that some GCMs also display this feature. However, in fact, the majority of GCMs show the opposite: higher CO2 levels are associated with weaker AMOC (e.g. Swingedow et al., 2007; Ma et al. 2021; Fortin et al., 2023), due to several processes (high latitude water cycle intensification, energy dissipation by eddies, etc.). Since the results presented*

*here show a very strong dependence of glacial inception to the AMOC state, I think that a more thorough discussion on how robust are the changes in AMOC can be considered in models (not specifically yours). You might have ideas why lower CO2 leads to weaker AMOC?*

The reviewer is perfectly right:  in all models, including all CLIMBER models, AMOC is weakening in response to abrupt or gradual $CO_2$ rise. However, this is a transient, centennial to millennial time scale response. In our paper, we performed experiments where the boundary conditions were kept constant for a sufficiently long time to reach an equilibrium state. Unfortunately, there are not many studies with GCMs where equilibrium AMOC response to different $CO_2$ levels was investigated. The first was by Stouffer and Manabe (2003), who found AMOC strengthening under doubling and quadrupling $CO_2$ and significant AMOC weakening under $CO_2$ halving in multimillennial simulations. Recently, Bonnan et al.

(2022) described AMOC response to instantaneous $CO_2$ quadrupling. Mots of runs were not long enough, and AMOC continues to evolve through the runs, but at least in one model (CESM1), AMOC is appreciably stronger under $CO_2$ quadrupling. For lower $CO_2$ concentration, there are two systematic studies (Galbraith and Lavergne, 2019; Oka et al., 2012) where it has been shown that global cooling causes a weakening of the AMOC. It was also shown in these studies that large ice sheets tend to stabilize AMOC, but this is not applicable to our study of glacial inception, where simulated ice sheets are rather small. Thus, the AMOC-$CO_2$ relationship simulated in CLIMBER-X is consistent with previous modeling results. There are likely several reasons for AMOC weakening under low $CO_2$, and one of them is related to a larger increase in AABW density compared to NADW. This, in turn, can be explained by enhanced sea ice formation in the Southern Ocean and salinity increase of AABW, the effect which has already been demonstrated in Ganopolski et al. (1998). As far as the reviewer's assumption about the "very strong dependence of glacial inception to the AMOC state" is concerned, as shown above, it is not correct – AMOC plays some, but definitely not decisive, role in glacial inception.
We have added some of this discussion to the revised paper.

**Specific comments**

*- l. 53. By construction, in the radiative scheme of CLIMBER-X there is a logarithm structure with respect to CO2, right? So there is no surprise here.*

Logarithmic dependence of radiative forcing on CO2 is not surprising, but it is not prescribed in the model. In CLIMBER-X the effect of CO2 (as well as water vapor) on radiation fluxes, is computed using the integral transmission function (see Appendix A8 in Willeit et al., 2022).

*- l. 64. "(iv)": be more specific please.*

Agreed. We have now added a list of specific processes that are better represented in CLIMBER-X compared to CLIMBER-2, e.g.: surface energy balance, precipitation, radiative transfer, sea ice dynamics, photosynthesis, vegetation dynamics.

*- l. 72. CLIMBER-X is really expected to reproduce the observed decadal variability? This is quite surprising given the model assumptions I guess.*

No, CLIMBER-X does not simulate **internal** decadal-scale variability by its design, but it can simulate the forced response at this time scale, such as response to volcanic eruptions and $CO_2$ changes (see Fig. 19 in Willeit et al., 2022). We added this to the revised manuscript.

*- l. 74. What is the vertical resolution of the oceanic model?*

Vertical ocean model resolution: 23 unequally spaced vertical layers, with a 10m top layer and layer thickness increasing with depth and reaching 500m at the ocean bottom (Willeit et al., 2022). We added this information to the revised manuscript.

*- l. 78. Is there any reference for SEMIX? I know that it has been used before but it could be useful to have*
*a reference here.*

SEMIX model description is given in accompanying paper by Willeit et al., 2024 which is currently under review in CPD. We hope, it will be published soon as CP paper. In the revised paper we explicitly added a reference to Willeit et al., 2024.

*- l. 84-85. Reference?*

We have added a reference to Willeit et al. (2022,2023).

*- l. 87. It would have been nice to have a map of the major biases (temperature / precip at least) or a reference here.*

Temperature biases are shown in Willeit et al. (2024) (Fig. B1). During the revision process in Willeit et al. (2024) we have now also added a figure showing the precipitation biases in the model (Fig. B3 in the
accepted version of Willeit et al. (2024)). In the revised paper we have added a reference to Willeit et al., (2024) when discussing the temperature biases and have also added a sentence about precipitation biases: "*As discussed in Willeit et al., (2024), the precipitation biases in CLIMBER-X are less problematic for the simulation of the surface mass balance of NH ice sheets.*"

*- l. 96-97. How do you conserve the water budget while using an acceleration factor? How do you*
*compute your freshwater flux? This is important since it will impact the AMOC in all your simulations.*

1) The conservation of the water budget in an Earth system models implies that the sum of water contents in the ocean, atmosphere, land and ice sheets remains constant. In CLIMBER-X, atmosphere-land and ice sheet components conserve water, but the ocean model (GOLDSTEIN) is based on the rigid-lid approximation, i.e. it does not include a prognostic equation for the ocean volume. Instead, the ocean
volume and global sea level are determined by the global ice volume and the volume of the ocean is regularly adjusted by scaling the thicknesses of the ocean layers below a depth of 1000m to match the actual ocean volume derived from sea level change (Willeit et al. 2022). This ensures that the global change in salinity and other tracers, which is important in the case of the interactive carbon cycle, are consistent with the simulated ice sheet volume. This procedure works the same way irrespectively of
whether acceleration of climate components is applied or not.

2) Freshwater into the ocean from ice sheets in the acceleration experiments is computed the same way as in nonaccelerated: annual freshwater flux into the oceans is computed as a sum of surface runoff routed in the ocean according to the ice sheet/land topography and calving (solid ice discharge), which is delivered to the nearest ocean grid cell.

3) As far as the impact of freshwater flux associated with the growth of ice sheet on AMOC is concerned, it is negligible. Indeed, in the simulations which is used to determine the critical insolation-CO2 relationship, growth of less than 10 meters in sea level equivalent occurs during at least several tenths of thousand years, which gives a net evaporation from the North Atlantic of less than 0.01 Sv. Such freshwater flux has a negligible effect on AMOC. The main difference in AMOC strength between experiments with low and high $CO_2$ is explained by the effect of temperature on AMOC.

*- l. 98-101. Do you need such a large acceleration factor since CLIMBER-X can perform 10 kyr a day right? Maybe it could have been nice to present a few additional experiments with no acceleration since it can affect your inception threshold.*

Yes, we need a significant acceleration of the climate component to perform the stability analysis of the
climate-cryosphere system in CO2-insolation phase space. In fact, 10 kyr per day is the performance of the climate component only (Willeit et al. 2022) and, in the case of interactive NH ice sheets (spatial resolution 30 km), CLIMBER-X is twice slower. Even though it is still much faster than GCM-based ESMs, one should not underestimate the amount of computations performed for this paper. To trace the stability diagram (Fig. 2) we performed experiments for 19 different orbital configurations. For each
orbital configuration, we performed, on average, about 25 simulations with different CO2 to trace the critical CO2 value with an accuracy of 5ppm. Since each run was 100,000 years long, the total simulation length is 50 million years (!). In the case of the acceleration factor=10, this is only the ice sheet model years, but without acceleration, it would also be climate model years. Taking into account the model performance of 5 hr per 1000 years and the fact that each model run uses 16 processors, such
simulations would require 4 million CPU hours. This is too expensive to produce just a single curve. Moreover, we have confidence that this can be done at least ten times cheaper, as demonstrated by Willeit et al. (2024).

In the revised paper we have added a sentence justifying the use of an acceleration factor of 10 based on the results in Willeit et al. (2024): "*Willeit et al. (2024) demonstrated that acceleration factors up to*
*~10 are acceptable and provide realistic results of the last glacial inception*."

*- l. 101. Why not using a properly spun-up ice sheet instead of using an ad-hoc vertical structure?*

The meaning of "proper spin-up" is unclear in the context of our ensemble of quasi-equilibrium experiments aimed at tracing the stability diagram of the climate-cryosphere system. For present-day Greenland, spin-up is usually performed by running the model through the previous glacial cycle, but in
the case of our study, there is nothing "previous" since our simulations are not related to real-time. Since the only ice sheet which is present in initial conditions is the Greenland Ice Sheet, we do not expect that the choice of initial temperature distribution in the Greenland ice sheet can affect in any way glacial inception, which is typically simulated 20 to 60 kyr after the beginning of the experiments and the GrIS has therefore enough timer to adjust to the fixed climate forcings (orbital and CO2).

*- Figure 1. It could have been useful to have indications for the different marine isotope stages in this graph (MIS 5,7,9 and 11).*

In this study (unlike Willeit et al., 2024), we did not model any specific (real) glacial inceptions. Instead, we just permutate different combinations of orbital parameters and $CO_2$ concentrations. Most of such combinations never materialized during the past 800 kyr. This is why referring in the paper to the real
MISs can be misinterpreted.

*- l. 156-158. You should perhaps temper this result since CLIMBER-X and CLIMBER-2 are not completely independent. In particular, they share a quite similar atmospheric model (although at a different spatial resolution).*

In fact, CLIMBER-2 and CLIMBER-X are completely different models. Apart from very different spatial resolutions, all components of CLIMBER-X differ significantly from those in CLIMBER-2 (even if they have the same names as, for example, SICOPOLIS). However, we fully agree with the reviewer that all modelling results, irrespective of model complexity, are model-dependent. In the revised paper we have rewritten the two sentences and we expanded the discussion on the robustness of the estimates of the parameters α and β:

*"The robustness of the estimated parameters of the critical insolation-$CO_2$ relationship is also supported by the theoretical analysis presented in Appendix A, where it is shown that for climate conditions similar to preindustrial, the value of α is about -80 $W/m^2$. Moreover, the value derived from the results presented by Abe-Ouchi et al. (2013) is -83 $W/m^2$. (Note that this number was not reported in Abe-Ouchi et al. (2013) but can be easily calculated from their Fig. 2). As was shown in Ganopolski et al. (2016), the values of β is reasonably well constrained by paleoclimate data since the critical insolation curve must pass between rather close insolation values corresponding to the end of MIS11 and present insolation. At the same time, paleo data provide no constraint on the value of α. This is why, in Talento and Ganopolski (2021), we used a very conservative approach by accepting as "valid" any α values from the range -150 to 0 $W/m^2$, i.e. we assumed relative uncertainties of up to 100%. The results of the present study strongly indicate that the uncertainty is much smaller, likely, not higher than 20%. Such a reduction of the uncertainty range would also significantly reduce the uncertainties in the projections of the timing of future glaciations for different anthropogenic $CO_2$ emissions."*

*- l. 261-269. Do these experiments display the same AMOC states as when using interactive ice sheets?*

Since AMOC is primarily controlled by $CO_2$ in our model, in these experiments, AMOC is similar, but not identical, to the corresponding experiments with interactive ice sheets.

*- l. 292-293. What about the role of shortwave radiation? The high CO2 simulation have a larger surface temperature, higher precipitation but also smaller shortwave radiation. I am a bit surprised to read here that the positive SMB is explained by larger precipitation rate. In my experience the extent of the accumulation area is primarily driven by temperature/shortwave (i.e. melt).*

We fully understand the reviewer's bewilderment. Indeed, it is generally recognized that the mass balance of the ice sheet is primarily controlled by insolation and temperature, while precipitation plays a secondary role. However, in Fig. 8, we do not compare cold climates with warm climates. Here, we compare climate conditions along our critical insolation-$CO_2$ line. The question we address here is how similar/different are climate conditions under which glacial inception occurs in the high $CO_2$ and low $CO_2$ worlds. Not surprisingly, in terms of summer temperature, climate conditions are rather similar because they all correspond to "cold" summers. But there are some interesting nuances, namely, that in the wet (high-$CO_2$ world), glacial inception can occur under summer temperatures similar to PI (but with high precipitation) in the areas of ice sheet nucleation, while in the dry (low-$CO_2$) world, glacial inception requires temperatures lower than PI because precipitation is also lower. Thus, these results do not contradict the notion that temperature and insolation are the primary drivers of glacial inception. In the revised manuscript we have completely rewritten this paragraph to make it clearer.

*- l. 336-338. Is this difference explained by AMOC differences, here again?*

The primary reason for the scattering of the individual points around the logarithmic curve in Fig. 2 is that the metric for orbital forcing, which we use in this and previous studies - maximum summer insolation at 65N - is a good, but not a perfect one. This is not surprising in the view that the effects of precession and obliquity on insolation differ both in space and in temporal dynamics. The "outlier" mentioned in the manuscript, corresponds to orbital parameters at 209 ka when obliquity was 24.3°, which is close to its maximum value during the past 800 kyr. At the same time, seven other points with a similar smx65 and which are clustering around the logarithmic curve in Fig. 2 correspond to times when obliquity was close to its average value or lower. This is consistent with the notion that obliquity may be slightly undervalued in the smx65 metric. Thus, one would expect that triggering of glacial inception under higher obliquity for the same smx65 would require a lower CO2, which is precisely what is seen in our modelling results. It is also worth mentioning that all real glacial inceptions of the last 800 years occurred during periods of low or medium obliquity.

In the revised paper this is now discussed in more detail in Sec. 3.3.

*- l. 410. Not necessarily, e.g. Harder et al. (2017).*

Apparently, the reviewer disagreed with our statement that "surface latent heat flux can be neglected during the melt season." This assumption is justified by the fact that during the short melt season in the considered region, latent heat flux is typically smaller by an order of magnitude than short-wave and long-wave radiation fluxes (e.g. Ettema et al., 2010). In principle, the effect of latent heat flux can be incorporated by modifying the parameter $\gamma$ in Eq. 5, but this parameter does not enter the final expression (Eq. 12) anyhow.

As far as the interesting results presented by Harder et al. (2017) are concerned, the situation with patchy snow cover at the end of the snow season for which their measurements were performed, obviously, is not applicable to large perennial snow fields, which are the precondition for glacial inception.

In the revised paper we have included the following sentence to justify why we neglect the latent heat flux: "*Surface latent heat flux can be neglected during the melt season, as it is an order of magnitude smaller than short-wave and long-wave radiation fluxes (e.g. Ettema et al., 2010).*"

*- l. 426. It is quite variable geographically and depends on continentality, seasonality of precipitation etc.*

Our assumption that "the duration of the melt season at the location of glacial inception is about two months" is applicable only to the region where glacial inception is simulated by our model, namely, the Canadian Arctic. Since present (more precisely, PI) climate conditions are rather close to the condition of glacial inception, the continentality, seasonality, precipitation, and other characteristics of climate in this area at the time of glacial inception must be similar to the known present ones. Note that the duration of the melt season only affects (and not significantly) the values of $k_S$ and $k_T$, which also do not enter the final expression (Eq. 12).

*- l. 431. I guess it comes from the temporal integral between the beginning and the end of the melt season? Why these are different for insolation and temperature then, if in both case the integral is over two month around the peak value?*

The reason is that although temperature and insolation have a similar seasonal evolution, they are described by different formulas. At 65°N, daily insolation $S(t)$ can be described with good accuracy as

$$S(t) = 0.5 S_{max} (1 + \cos \omega t) ,$$

while surface air temperature is given by the formula

$$T(\tau) = A\cos\omega\tau + B = T_{max} + A(\cos\omega\tau - 1),$$

where $\omega = 2\pi/365$, $t$ is time counted from the summer solstice, $\tau$ is time counted from the date of temperature maximum which lags summer solstice by about month ($t-\tau \approx 30$ days), $A$ is the amplitude of seasonal temperature variations and $T_{max}$ maximum surface air temperature. This is why, when integrating these two characteristics during the melt season, the relationship between average insolation and maximum insolation, and between averaged temperature and maximum temperature, are different. However, the values $k_S$ and $k_T$ are not very sensitive to the choice of the duration of the melt season, and as it was noted above, these parameters do not enter the final formula (Eq. 12).
To clarify this, in the revised paper we have added a Figure A1 illustrating a typical seasonal evolution of temperature and insolation under representative glacial inception conditions.

Bonan, D.B., Thompson, A.F., Newsom, E.R., Sun, S. and Rugenstein, M.: Transient and equilibrium responses of the Atlantic overturning circulation to warming in coupled climate models: The role of temperature and salinity. Journal of Climate, 35, 5173-5193, 2022.

Ettema, J., van den Broeke, M. R., van Meijgaard, E., and van de Berg, W. J.: Climate of the Greenland ice sheet
using a high-resolution climate model – Part 2: Near-surface climate and energy balance, The Cryosphere, 4, 529–544, https://doi.org/10.5194/tc-4-529-2010, 2010.

Galbraith, E., and de Lavergne, C.: Response of a comprehensive climate model to a broad range of external forcings: relevance for deep ocean ventilation and the development of late Cenozoic ice ages. Climate Dynamics, 52, 653-679, 2019.

Ganopolski, A., S. Rahmstorf, V. Petoukhov, and M. Claussen: Simulation of modern and glacial climates with a coupled global model of intermediate complexity, Nature, 391, 351-356, 1998.

Oka, A., Hasumi, H. and Abe-Ouchi, A.: The thermal threshold of the Atlantic meridional overturning circulation and its control by wind stress forcing during glacial climate, Geophys. Res. Lett., 39, L09709, 2012.

Stouffer, R.J. and Manabe, S.: Equilibrium response of thermohaline circulation to large changes in atmospheric
CO2 concentration. Climate Dynamics, 20, 759-773, 2003.

Talento, S., Ganopolski, A.: Reduced-complexity model for the impact of anthropogenic CO2 emissions on future glacial cycles, Earth System Dynamics, 12 , 1275-1293, 2021.

Willeit, M., Ganopolski, A., Robinson, A. and Edwards, N.R.: The Earth system model CLIMBER-X v1. 0–Part 1: Climate model description and validation. Geoscientific Model Development, 15(14), 5905-5948, 2022.

Willeit, M., Calov, R., Talento, S., Greve, R., Bernales, J., Klemann, V., Bagge, M., and Ganopolski, A. Glacial inception through rapid ice area increase driven by albedo and vegetation feedbacks, EGUsphere, https://doi.org/10.5194/egusphere-2023-1462, 2024.

**Referee#2**

**Main comments**

*1 - This study is presented as a theoretical work aimed at better understanding the relative role of insolation and CO2 in the triggering of glacial inception, with the objective to better predict "future glaciations and the effect that anthropogenic CO2 emissions might have on them". This paper is indeed fully relevant in this respect. It complements a previous study (Ganopolski et al. 2016) made with a simpler model. But in contrast to this previous paper based on CLIMBER-2, CLIMBER-X appears to be a*

*rather new model. In particular, it is not clear how well it can simulate the actual glacial inceptions observed during the Quaternary. The authors are citing a preprint (Willeit et al. 2024) concerning the last glacial inception, but it is difficult to evaluate how well this new model configuration behaves on the other inceptions. I would appreciate some discussions or comments on this point, for instance by building on improvements made versus CLIMBER-2, or by discussing a bit more the simulations corresponding to*

*the actual last 4 inceptions, among the 19 simulations performed. It would strengthen the paper to put these results against observations, even if they are not fully comparable.*

The reviewer is perfectly right – CLIMBER-X is a brand-new model, superior to CLIMBER-2 in all respects. CLIMBER-X was extensively evaluated against observational data and results of more complex models. The successful simulation of the past glacial inception (Willeit et al. 2024) is also part of such validation.

In addition, similar to Ganopolski et al. (2016), the model's parameters were selected to meet two clearly defined empirical constraints: the model should not simulate glacial inception at the end of the Holocene but simulate glacial inception at the end of MIS11. Since the critical CO2-insolation relationship obtained in this study is very similar to that in Ganopolski et al. (2016), the fig 3b from the 2016 paper shows that CLIMBER-X should simulate all previous glacial inceptions since they are located below the critical CO2-

Insolation line. The question of how realistically the model can simulate previous glacial inceptions is not possible to address because prior to MIS5, the only empirical information which could be used to compare with modelling results is the global sea level or ice volume (e.g. Elderfield et al. 2012; Grant. et al. 2014; Rohling et al., 2014; Spratt and Lisiecki, 2016; Waelbroeck et al. 2002), but these reconstructions strongly disagree with each other, in particular during glacial inceptions. This fact makes simulations of previous glacial inception of little use for model validation.

*2 - It is not discussed if climate sensitivity is different, or very similar, in CLIMBER-X versus CLIMBER-2 (I suspect the latter), if the radiative code is identical or not. Such information would be critical to assess such statements as (line 155): "values for alpha and beta might not be strongly model-dependent". This would likely not hold with very different climate sensitivity to CO2 forcing. A few lines of information on*

*CLIMBER-X in this respect would be useful.*

Although CLIMBER-2 and CLIMBER-X are very different models, they do have similar equilibrium climate sensitivities close to the "IPCC best guess" of 3K. The reviewer is right - climate sensitivity should affect the slope of the critical insolation-CO2 relationship. In any case, modelling results are always model-dependent.

In the revised paper we have added the information on equilibrium climate sensitivity in CLIMBER-X and expanded the discussion on the robustness of the estimates of the parameters α and β:
"*The robustness of the estimated parameters of the critical insolation-$CO_2$ relationship is also supported by the theoretical analysis presented in Appendix A, where it is shown that for climate conditions similar to preindustrial, the value of α is about -80 $W/m^2$. Moreover, the value derived from the results presented*

*by Abe-Ouchi et al. (2013) is -83 $W/m^2$. (Note that this number was not reported in Abe-Ouchi et al.*

*(2013) but can be easily calculated from their Fig. 2). As was shown in Ganopolski et al. (2016), the values of β is reasonably well constrained by paleoclimate data since the critical insolation curve must pass between rather close insolation values corresponding to the end of MIS11 and present insolation. At the same time, paleo data provide no constraint on the value of α. This is why, in Talento and Ganopolski (2021), we used a very conservative approach by accepting as "valid" any α values from the range -150 to 0 W/m², i.e. we assumed relative uncertainties of up to 100%. The results of the present study strongly indicate that the uncertainty is much smaller, likely, not higher than 20%. Such a reduction of the uncertainty range would also significantly reduce the uncertainties in the projections of the timing of future glaciations for different anthropogenic $CO_2$ emissions."*

*3a - Line 89-90: biais correction on temperature. It is all right to use such a procedure, but it would be necessary to have some discussion on possible impacts on the final results. The hidden assumption is that the biaises should remain constant, whatever the climate and the ice-sheet evolution. Is it realistic?*

The assumption about constant temperature biases is not so problematic for the given study, where we
are only interested in the initial phase of ice sheet growth. Obviously, such an assumption would be much less justified for modelling the entire glacial cycles. As shown in Fig. 8, glacial inception in different experiments happened under conditions that are rather similar to modern summer temperature conditions. This is not surprising since PI climate was already very close to glacial inception. Thus, prior to the appearance of large ice sheets, summer temperature biases are not expected to be very different for
different combinations of orbital parameters and CO2. Since glacial inception is diagnosed by the growing of additional ice by less than 10 msl, such ice sheets are by order of magnitude smaller (both in area and volume) than the LGM-size ice sheets and their impact on atmospheric circulation is expected to be rather small. Since the main cause for summer temperature biases over North America is a bias in simulated atmospheric circulation, there is no reason to expect that a small Northern American ice sheet
would have an appreciable influence on climate biases. This issue is now discussed in more detail in the revised version of Willeit et al., 2024.

*3b - There is no mention of "biais correction" on precipitation. I read through Willeit et al. (2024) Appendix B but found no information on this point. I had a look at Willeit et al. (2022) but could not really evaluate precipitations at high latitude. Are precipitations from SESAM good enough?*

Present-day annual precipitation simulated by CLIMBER-X over the region where ice sheets were growing during glacial inceptions is in reasonable agreement with reality (see figure below), and typical biases do not exceed 200 mm/yr. At the same time, the effect of +1°C summer temperature biases on annual snowmelt can be estimated using the classical PDD approach to be 200-500 mm/yr (the low bound corresponds to the melt season duration of two months and parameter $\alpha$=3 mm/(°C day), while the
upper corresponds to the melt season duration of three months and $\alpha$=5). Since simulated summer temperature biases in this region are about 3-5°C (Fig. 2a in Willeit et al., 2024), temperature biases are much more important than precipitation biases. This is why we only corrected the temperature.
The figure below has not been included in the revised version of Willeit et al. (2024) as Fig. B3 and the relative impact of precipitation versus temperature biases is also discussed in Willeit et al. (2024). In this
paper we have now also added a sentence about precipitation biases: "*As discussed in Willeit et al., (2024), the precipitation biases in CLIMBER-X are less problematic for the simulation of the surface mass balance of NH ice sheets.*"

[Figure]

**Fig. 1.** Simulated present-day annual precipitation (left), reanalysis data (middle), difference between model and data (right).

*These two points may be critical for instance in the discussion of Hsmx65_LCO2_Fixice (cold) versus*
*Lsmx65_HCO2_Fixice (snowy).*

The main difference between Hsmx65_LCO2 and Lsmx65_HCO2 is in temperatures, not in precipitation. In North America, summer temperature differences between these two experiments are rather small (fig. 8) and, as a result, ice sheet configurations are essentially identical (Fig. 4). In Hsmx65_LCO2 Scandinavia is colder (partly due to a weak AMOC) and as a result, an ice sheet is growing over
Scandinavia in Hsmx65_LCO2 although precipitation in these experiments is lower here than in the Lsmx65_HCO2 run (Fig. 8d and f). In the revised paper this is discussed in more detail in Sec. 3.1.

**Minor comments**

*4 - (line 96): "sea level (which affects land-sea mask)". I guess this concerns the atmospheric model, but not the ocean model (bathymetry)? Or does the ocean have a bathymetry which adapt to ice-sheet*
*induced changes?*

Yes, the ocean bathymetry, as well as the land elevation above sea level and river routing scheme, are updated every 10 years : "Changes in sea level, and therefore ocean volume, are additionally accounted for by scaling the thicknesses of the ocean layers below a depth of 1000m to match the actual ocean volume derived from the high-resolution topography and provided as input to the ocean model. Total
tracer inventories in the ocean are conserved in this process". (Willeit et al. 2022, p. 5912)

*5 - (line 170, legend Fig 2) CLIMEBR -> CLIMBER*

Fixed, thanks.

*6 - (line 241) understating -> understanding*

Fixed, thanks

Elderfield, H. et al. Evolution of ocean temperature and ice volume through the mid-Pleistocene climate transition. Science 337, 704–709, 2012.

Grant, K.M. et al. : Sea-level variability over five glacial cycles. Nature communications, 5, 5076, 2014.

Rohling, E.J., et al.: Sea-level and deep-sea-temperature variability over the past 5.3 million years. Nature, 508, 477–482, 2014.

Spratt, R. M. and Lisiecki, L. E.: A Late Pleistocene sea level stack, Clim. Past., 12, 1079–1092, 2016.

Talento, S., Ganopolski, A.: Reduced-complexity model for the impact of anthropogenic CO2 emissions on future glacial cycles, Earth System Dynamics, 12 , 1275-1293, 2021.

Waelbroeck, C., et al.: Sea level and deep water temperature changes derived from benthic foraminifera isotopic records, Quaternary Sci. Rev., 21, 295–305, 2002.

Willeit, M., Ganopolski, A., Robinson, A. and Edwards, N.R.: The Earth system model CLIMBER-X v1. 0–Part 1: Climate model description and validation. Geoscientific Model Development, 15(14), 5905-5948, 2022.

Willeit, M., Calov, R., Talento, S., Greve, R., Bernales, J., Klemann, V., Bagge, M., and Ganopolski, A. Glacial inception through rapid ice area increase driven by albedo and vegetation feedbacks, EGUsphere, https://doi.org/10.5194/egusphere-2023-1462, 2024.

---

## Author Response (AR2)

**Referee#1**

*I only have a few minor comment about water conservation and freshwater flux. I understand that you conserve the volume of water by adjusting the ocean volume to account for ice sheet volume change. At the same time you route the surface runoff and calving to compute the FWF. However, for one given climate year you have 10 ice sheet model years, meaning that the FWF that arrives to the ocean every year corresponds to 10 times the annual flux. I think you should explicitly mention this fact in the experimental setup section.*

We now better understand the reviewer's concern about the potential influence of the acceleration technique on our model results. Indeed, if we were to calculate global ocean volume change by integrating surface freshwater fluxes, then a 10-fold acceleration would require us to aggregate these fluxes over 10 years, which would have a significant impact on model results. However, since the ocean model we use (GOLDSTEIN) is based on the rigid-lid approximation we always enforce that the net annual global surface freshwater flux is zero. In the case of interactive ice sheets, we still enforce the global annual surface freshwater flux to be zero, but then we adjust the ocean volume each year to match the change in global ice volume. This is done, as we explained in the first response, by scaling the thicknesses of the ocean layers below 1000 m depth to match the actual ocean volume derived from sea level change (Willeit et al. 2022). In doing that, we also recalculate salinity and other tracers to enforce their conservation. Since our ocean volume change is not driven by surface freshwater fluxes, we calculate them in the same way irrespective of whether we use acceleration or not. Namely, the surface freshwater flux into the ocean is equal to the sum of daily values of precipitation - evaporation, runoff from ice-free land, surface runoff from the ice sheets, plus annual mean values of calving and basal melt from the ice sheets, calculated by the ice sheet model every year, but only passed to the ocean model in the year in which the climate model is updated (every ten years in the case of the accelerated run). Of course, such an approach introduces some inconsistency between the freshwater fluxes computed in the atmospheric and ice sheet models and the fluxes entering the ocean model. However, these inconsistencies are less than 1% of the typical net freshwater fluxes to the ocean, which is of course much smaller than typical model errors in simulated components of the freshwater forcing.

*Also, l.137, remove "easily".*

Removed.